# Molecular Characterization of Two Totiviruses from the Commensal Yeast *Geotrichum candidum*

**DOI:** 10.3390/v15112150

**Published:** 2023-10-25

**Authors:** Mahmoud E. Khalifa, Robin M. MacDiarmid

**Affiliations:** 1Botany and Microbiology Department, Faculty of Science, Damietta University, Damietta 34517, Egypt; mahmoud.khalifa@du.edu.eg; 2The New Zealand Institute for Plant and Food Research Limited, Auckland 1025, New Zealand; 3School of Biological Sciences, The University of Auckland, Auckland 1010, New Zealand

**Keywords:** mycoviruses, dsRNA, *Totivirus*, *Geotrichum*, high-throughput sequencing, siRNA, RNA interference, usRNA

## Abstract

Mycoviruses can infect many of the major taxa of fungi including yeasts. Mycoviruses in the yeast fungus *Geotrichum candidum* are not well studied with only three *G. candidum*-associated viral species characterized to date, all of which belong to the *Totiviridae* genus *Totivirus*. In this study, we report the molecular characteristics of another two totiviruses co-infecting isolate Gc6 of *G. candidum*. The two totiviruses were tentatively named Geotrichum candidum totivirus 2 isolate Gc6 (GcTV2-Gc6) and Geotrichum candidum totivirus 4 isolate Gc6 (GcTV4-Gc6). Both viruses have the typical genome organization of totiviruses comprising two ORFs encoding capsid protein (CP) and RNA-dependent RNA polymerase (RdRp) at the N and C termini, respectively. The genomes of GcTV2-Gc6 and GcTV4-Gc6 are 4592 and 4530 bp long, respectively. Both viruses contain the—frameshifting elements and their proteins could be expressed as a single fusion protein. GcTV2-Gc6 is closely related to a totivirus isolated from the same host whereas GcTV4-Gc6 is related to insect-associated totiviruses. The phylogenetic analysis indicated that GcTV2-Gc6 and GcTV4-Gc6 belong to two different sister clades, I-A and I-B, respectively. It is interesting that all viruses identified from *G. candidum* belong to the genus *Totivirus*; however, this might be due to the lack of research reporting the characterization of mycoviruses from this fungal host. It is possible that the RNA interference (RNAi) mechanism cannot actively suppress totivirus accumulation in *G. candidum* Gc6.

## 1. Introduction

The ascomycetous yeast *Geotrichum candidum* is a common, dimorphic member of the Saccharomycetes class [1]. It is known to infect plants and cause post-harvest rot in a variety of fruit and vegetable crops [2]. It supports the growth of *Penicillium camemberti* through the metabolism of lactate and suppresses the growth of undesired organisms, and hence, is used by the commercial cheesemaking industry as a ripening culture [3,4]. Moreover, it is a commensal parasite able to infect human skin, respiratory tracts, and gastrointestinal tracts [5,6].

Mycoviruses, viruses that infect and replicate in fungi, have been reported in several fungi belonging to all phyla of true fungi [7] as well as oomycetes [8]. Although mycovirus infections are mostly associated with symptomless infections, they may impact their fungal hosts in several ways [9,10,11,12]. Mycoviruses with three genome types (single-stranded RNA (ssRNA), double-stranded RNA (dsRNA), and ssDNA) have been reported with ssDNA mycoviruses being less common [7,9,13,14]. According to the International Committee on the Taxonomy of Viruses (ICTV), dsRNA mycoviruses are classified within the families *Alternaviridae*, *Amalgaviridae*, *Birnaviridae*, *Chrysoviridae*, *Curvulaviridae*, *Megabirnaviridae*, *Partitiviridae*, *Polymycoviridae*, *Quadriviridae*, *Spinareoviridae*, and *Totiviridae*, and the genus *Botybirnavirus* (https://ictv.global/taxonomy, accessed on 10 September 2023). Members of *Totiviridae* family are classified into five genera: *Giardiavirus*, *Trichomonasvirus*, *Leishmaniavirus*, *Victorivirus*, and *Totivirus* [15]. The known host range of *Totiviridae* members and totivirus-like genomes has expanded over recent years, with the increased utilization of next-generation sequencing technologies, to include insects, arthropods, fish, crab, and plants [16,17,18,19,20]. Historically, virus-like particles (VLPs) and dsRNAs have been detected in *G. candidum* without further biological or molecular identification of these associated VLPs or dsRNAs [21,22]. Recently, four totiviruses were detected and characterized from *G. candidum* isolates from Pakistan [23]. The current study reports the characterization of two mycoviruses from a different isolate of the same fungus. Interestingly, the two mycoviruses were also classified into the *Totiviridae* family, indicating that totiviruses are common in *G. candidum*.

## 2. Materials and Methods

### 2.1. Isolation, Maintenance, and Identification of Isolate Gc6

Isolate Gc6 was isolated along with other fungal species from a soil sample collected from Amberely, New Zealand, and grown on potato dextrose agar (PDA) using the soil-plate method [24]. A pure culture was obtained, and DNA was extracted using a Zymo DNA Fungal/Bacterial Miniprep Kit as described by the manufacturer. The identity of isolate Gc6 was determined by amplifying and sequencing the non-coding internal transcribed spacer (ITS) region of the fungal ribosomal DNA (rDNA) using the primer pair ITS4/ITS5 [25]. Throughout the course of this study, isolate Gc6 was maintained and cultured on PDA plates.

### 2.2. Purification and Visualization of dsRNA

Potato dextrose broth (PDB) media was inoculated with freshly grown fungal mycelial plugs and incubated at 25 °C for 5 days. Two grams of fungal mycelia was harvested and used to survey for the presence of dsRNA using a method based on the selective binding of dsRNA to cellulose powder in the presence of 16.5% ethanol as previously described [26]. Purified dsRNA was treated with DNase and RNase in a high salt buffer [27], separated on an ethidium bromide-prestained 1% (*w*/*v*) agarose gel in Tris–acetate EDTA (TAE) buffer, and visualized on a UV transilluminator.

### 2.3. Sequencing, Bioinformatics, and Sequence and Phylogenetic Analyses of dsRNA

The purified dsRNA fraction was used as a template for a reverse transcription polymerase chain reaction (RT-PCR) to generate random cDNA as described by Khalifa and Pearson [28]. The RT-PCR products were purified using a Gel/PCR DNA purification kit (Geneaid Biotech Ltd., New Taipei City, Taiwan) and sequenced using an Illumina HiSeq2000 at Macrogen Inc. (Seoul, Republic of Korea). Illumina short reads were quality checked using FastQC and based on the quality check reports, Illumina reads were filtered and trimmed based on quality. The reads were then de novo assembled using Geneious Prime 2022.1.1 software (Biomatters, New Zealand) set to medium sensitivity and default parameters. The assembled contigs were identified using Blastx against the GenBank non-redundant (nr) database. To determine the sequence of dsRNA termini and obtain full-length sequences of the viral contigs, the T4L adapter (5′-PO_4_-CCCGTCGTTTGCTGGCTCTTT-NH_2_-3′) was ligated to the 3′ end of the dsRNAs using T4 RNA ligase enzyme. The T4L-ligated dsRNAs were purified using a PCR clean-up and gel extraction kit (GeneDireX, Taoyuan City, Taiwan) following the manufacturer’s instructions, and used as a template for RT-PCR amplification of the terminal sequences using the T4LC primer (5′-AAAGAGCCAGCAAACGACGGG-3′) together with dsRNA specific primers designed based on the sequences obtained by Illumina sequencing (Appendix A). RT-PCR products corresponding to the terminal sequences were purified and Sanger sequenced by Macrogen Inc. (Seoul, Republic of Korea).

Open reading frames (ORFs) were detected using the ORF finder tool (https://www.ncbi.nlm.nih.gov/orffinder/, accessed on 1 May 2023). Viral protein sequences were aligned using MAFFT [29] in order to identify conserved motifs. For the phylogenetic analysis, related protein sequences were retrieved from the NCBI database (accessed in April 2023) and aligned using MAFFT. Based on the obtained alignments, MEGA-X 10.2.5 software [30] was used to construct maximum-likelihood (ML) phylogenetic trees using the best fit models. A le–Gascuel model with a gamma distribution and some sites set to be evolutionary invariable (LG + *G* + *I*) and General Reverse Transcriptase + Freq with a gamma distribution (rtREV + *G* + *F*) model with 1000 bootstrap replicates were used for RNA-dependent RNA polymerase (RdRp) and capsid protein (CP) ML phylogenetic trees construction, respectively.

### 2.4. Small RNA (sRNA) Purification and Sequencing

*G. candidum* Gc6 was grown on cellophane-covered PDA Petri dishes for 5 days at 25 °C. The sRNA fraction was purified from grown mycelia using a mirPremier™ microRNA Isolation Kit (Sigma-Aldrich, Burlington, MA, USA). Purified microRNAs were run on an agarose gel to confirm their integrity and their concentration and purity were determined using a NanoDrop spectrophotometer. An aliquot of 20 µL of purified microRNAs at a concentration of 369 ng/µL was sent to Marogen Inc. (Seoul, Republic of Korea) for library construction and sequencing. A library of sRNA was prepared using a Truseq small RNA kit and sequenced using Illumina sequencing on one lane of a HiSeq2000 platform with a read length of 50 bp SR. For library preparation, RNA 3′ adapter (RA3) and RNA 5′ adapter (RA5) were ligated to RNAs using a T4 RNA ligase enzyme. Adapter-ligated libraries were reverse transcribed using SuperScript II reverse transcriptase and 2 primers that anneal to the adapter ends. Following amplification, the library was run on a high-sensitivity DNA chip for a quality check (size, purity, and concentration). cDNA constructs were gel purified in preparation for subsequent cluster generation. Constructs representing different sRNA species such as miRNAs, siRNAs, and Piwi-interacting RNAs were normalized to 2 nM using Tris-HCl 10 mM, pH 8.5, and sequenced. Sequencing data were obtained in FASTQ format. The obtained Illumina reads were trimmed before being mapped against the full sequences of dsRNAs obtained from *G. candidum* Gc6 using Bowtie2 with default parameters [31]. As a control, the obtained reads were also mapped against the *G. candidum* reference genome LMA-244_clib (GenBank: GCA_013365045.1).

## 3. Results and Discussion

The taxonomy of fungal viruses has grown extensively in the past few years due to the increasing interest in mycoviruses and their associated effects on the one hand, and with the practicability of high-throughput sequencing (HGS) technologies and bioinformatic software on another hand [32]. Below, we report the molecular characteristics of two dsRNA mycoviruses from the ascomycete *G. candidum*.

### 3.1. Identity of the Host Fungus

Soil is a rich source of different types of microorganisms that may be infected with viruses. Since these viruses affect the metabolic activity of their hosts, they play a pivotal role in the functionality of the soil ecosystems [33]. Fungi are one of the main soil microbial components that can harbor a wide range of viruses causing variable effects on their hosts starting from symptomless to severely debilitating infections [7]. Isolate Gc6 was isolated from a soil sample on PDA using standard isolation methods. A pure culture was obtained (Figure 1a), and the fungal identity was determined by sequencing the ITS region. The nucleotide sequence of the amplified fragment shared nucleotide (nt) sequence identities greater than 98% with the corresponding ITS regions of several *G. candidum* isolates (Appendix A). Therefore, Gc6 was considered an isolate of *G. candidum* fungus.

### 3.2. Presence and Sequencing of dsRNA Associated with G. candidum Isolate Gc6

Electrophoretic analysis of the dsRNA purified from *G. candidum* Gc6 revealed the presence of a single nucleic acid band with an estimated molecular size of about 4.5 kb. This band resisted DNase treatment and RNase treatment in a high salt buffer, confirming its double-stranded nature (Figure 1b). Yeast fungi have been long known as hosts of mycoviruses since the discovery of a killer phenotype associated with *Saccharomyces*-virus killer systems [34]. In *G. candidum*, uncharacterized dsRNAs and virus-like particles have been discovered in the past [21,22]. Recently, three species belonging to the family *Totiviridae* have been characterized at the molecular level [23].

The purified dsRNA from *G. candidum* isolate Gc6 was subjected to random cDNA synthesis and sequencing, and its composition and identity were determined. De novo assembly of the Illumina short reads resulted in the creation of three long contiguous sequences 4519, 2447 and 743 nt in length that shared identities with totiviral sequences based upon initial Blastx analysis. The assembly was completed and validated, and the full-length sequences of the Gc6 dsRNAs were obtained using RACE-sequencing. The Gc6-dsRNAs were found to be 4592 (dsRNA1: contig 1) and 4530 (dsRNA2: contig 2 and 3) nt in length. Blastx searches of dsRNA1 and dsRNA2 full-length sequences revealed up to 94.15 and 89.24% identities compared to RdRp and 97.50% and 91.69% identities compared to CP sequences of viruses in the family *Totiviridae*, respectively (Appendix A). Therefore, the apparently single dsRNA segment purified from isolate Gc6 consists of two co-migrating segments representing the genomes of two totiviruses. The closest fungal virus to dsRNA1 was Geotrichum candidum totivirus 2 (GcTV2) and therefore, dsRNA1 was considered as an isolate of the same virus (GcTV2-Gc6). dsRNA2 shared the highest identity with an insect totivirus, and the closest fungal totivirus shared RdRp and CP aa sequence identities of 48.05 and 41.86% with dsRNA2. Hence, dsRNA2 was given a new tentative name, Geotrichum candidum totivirus 4 (GcTV4-Gc6). The sequences of GcTV2-Gc6 and GcTV4-Gc6 were deposited to GenBank under accession numbers OR250782 and OR250783, respectively. Infection of a single fungal isolate with multiple related or unrelated mycoviruses has been previously reported in several fungal species [23,28,35,36,37,38,39,40,41,42]. Mycoviruses previously discovered from *G. candidum* as well as those described in the current study were found to cause co-infections in one host isolate.

### 3.3. Genome Organization and Analysis of Geotrichum candidum Totiviruses

The *Totiviridae* family includes viruses with a single linear, uncapped dsRNA genome segment of approximately 4.6–7.0 kilobase pair (kbp) in length [15]. The genomes of its members consist of two large, sometimes overlapping, ORFs with the potential to encode major CPs of about 70–100 kilodalton (kDa) (5′ ORF) and an RdRp (3′ ORF). The GcTV2-Gc6 genome has a GC content of 45.5% and has the potential to code for two long open ORFs that are separated by a 326 nt long intergenic region (IR). ORF1 in frame three is 2043 nt long and is located following a 29 nt long 5′ untranslated region (UTR). It starts at nt position 30, ends at an UAA termination codon at nt position 2072, and encodes a 680 amino acid (aa) long protein with an estimated molecular mass of 76.6 kDa. ORF2 in frame two precedes a 37 nt long 3′ UTR and has a length of 2157 nt. It starts at an AUG codon (nt position 2399), terminates at an UAA codon (nt position 4555), and encodes a 718 aa long protein with an estimated molecular weight of 82.06 kDa.

The GcTV4-Gc6 genome has a GC content of 42.6% and consists of 2097 (ORF1) and 2373 (ORF2) nt long ORFs with the potential to encode proteins with sizes of 698 and 790 aa and estimated molecular weights of 78.6 and 90.11 kDa, respectively. ORF1 and ORF2 start at nt positions 6 and 2123 with an AUG codon and end at nt positions 2102 and 4495, respectively. The two ORFs are separated by a 20 nt long IR and the genome has 5 and 35 nt long UTRs at the 5′ and 3′ termini, respectively. The genome organization of GcTV2-Gc6 and GcTV4-Gc6 is similar to genomes of other *G. candidum* totiviruses [23] and is typical for those of yeast totiviruses coding for two ORFs [43].

Blastp searches of putative proteins translated from ORF1 and ORF2 (for both GcTV2-Gc6 and GcTV4-Gc6) against the nr protein sequences database returned several hits for CP and RdRp of totiviruses, respectively. In most totiviruses, ORF1 and ORF2 are expressed as a single fusion protein due to the existence of a −1 ribosomal frameshifting mechanism. A slippery site where the frameshifting occurs is required near the stop codon of ORF1 and represented by the canonical sequence XXXYYYZ where X is A/G/C/U, Y is A/U, and Z is A/C/U [44]. A pseudoknot secondary structure preceding the slippery site is required to pause the ribosome and increase the efficiency of frameshifting [45]. Putative slippery sites were found at the 3′ terminus of ORF1 in both viruses. This was represented by the nt stretches GGGUUUA (nt positions 1958–1964) and UUUUUUA (nt positions 1943–1949) for GcTV2-Gc6 and GcTV4-Gc6, respectively. The slippery site in GcTV2-Gc6 is identical to that of the Saccharomyces cerevisiae virus L-A (ScV-L-A) [46]. In GcTV2-Gc6, the slippery site and an H-type pseudoknot at nt positions 1969–2023 are separated by a 4 nt spacer (Figure 1c). Similarly, the slippery site of GcTV4-Gc6 was followed by a 27 nt long spacer and an H-type pseudoknot at nt positions 1977–1997 (Figure 1c). Although the presence of −1 frameshifting signature does not necessarily reflect its functionality [47], the CP and RdRp genes of GcTV2-Gc6 and GcTV4-Gc6 could be expressed as fusion proteins with sizes of 1508 and 1496 aa, respectively.

The RNA molecules synthesized by totiviral RdRps are not capped at their 5′ termini which makes them unrecognizable by their host translation machinery [48,49]. Therefore, it is established that totiviruses such as ScV-L-A and ScV-L-BC can de-cap their host’s mRNAs and transfer their caps to the totiviral transcripts using a unique cap-snatching mechanism that requires a certain histidine aa residue in the capsid protein [50,51]. Multiple aa sequence alignments of GcTV2-Gc6 and GcTV4-Gc6 CPs with other totiviral CP sequences revealed the presence of the conserved histidine aa at positions 154 and 158 of the translated protein (Figure 2a), respectively. This is analogous to the histidine residues found at aa positions 154, 15,6 and 159 of CPs of ScV-L-A [50], ScV-L-BC [51], red clover powdery mildew-associated totiviruses (RPaTVs) [52], and Trichoderma koningiopsis totivirus 1 (TkTV1) [53], respectively. Moreover, the aa sequence alignment between RdRps of GcTV2-Gc6, GcTV4-Gc6, and other totiviruses verified the presence of an RT-like superfamily domain (pfam02123) with eight conserved motifs (Figure 2b) including the GDD motif which is highly conserved among RdRp sequences of dsRNA viruses [54].

### 3.4. Phylogenetic Analysis

The phylogenetic structure of the genus *Totivirus* was determined, and members were found distributed in two major groups (groups I and II). Members in the first clade (group I) were further separated into four sister clades (I-A to I-D) [52]. The current phylogenetic status of the *Totivirus* genus has been re-determined in this study based on the increasing diversity of the discovered totiviral genomes, partially due to the advent of HGS. The phylogenetic trees constructed based on multiple alignments of RdRp and CP genes (Figure 3 and Figure 4) revealed that two additional sister clades (I-E and I-F) were formed in group I. The appearance of these two sister clades is taxonomically sensible since subclade I-E includes plant totiviruses exclusively and the sister clade I-A comprises totiviruses from a range of hosts including plant totiviruses. Moreover, subclade I-F that hosts totiviral members from algae is more related to subclade I-B where a related algal totivirus is found. Fungal totiviruses are distributed among subclades I-A to I-D among which, subclade I-D has no totiviruses from other hosts. *G. candidum* totiviruses are distributed among subclades I-A and I-B [23] which have totiviruses from multiple hosts including insects.

GcTV2-Gc6 was placed together with GcTV2-E1 [23] in subclade I-A and is most closely related to yeast and insect totiviruses. The closest viruses to GcTV4-Gc6 were isolated from insects (accession numbers ON812606 and ON812795). The existence of related fungal and insect totiviruses has previously been reported, reflecting the existence of common ancestors and possible horizontal virus transmission [52,53]. Previous reports suggested that unknown totiviruses from a wide range of different hosts may exist [52] and this was supported by the presence of closely related fungal and insect totiviruses such as TkTV1, Wuhan insect virus 26 (WIV26), and WIV27 [53]. As shown in Figure 3, this suggestion was confirmed, and it is believed that the diversity of totiviruses will expand further.

### 3.5. Are G. candidum Totiviruses Targets for the Host’s RNA Interference (RNAi) System?

Viral infections are repelled by their host’s defense machinery through RNAi which is a sequence-specific post-transcriptional gene silencing induced by dsRNA [55]. Viral dsRNA is common during the replication of all viral genome types. It is the replicative form during the replication of ssRNA viruses or duplex regions in complementary stretches of their genomes as well as in the transcripts of DNA viruses. Since the nature of dsRNA viral genomes is double-stranded, it is likely that the viral genome itself has the ability to trigger RNAi [55,56,57]. Therefore, the accumulation of *G. candidum* totivirus-derived short interfering RNAs (siRNAs) within *G. candidum* Gc6 implies that the host’s RNAi machinery is actively attempting to suppress viral replication and accumulation. A total small RNA (sRNA) fraction was purified from *G. candidum* Gc6, sequenced, and mapped against the genomes of GcTV2-Gc6 and GcTV4-Gc6 to obtain the viral-derived sRNAs. As a control to confirm the validity of the sRNA sequencing, sRNAs were also mapped against the LMA-244_clib reference genome of the *G. candidum* fungus. A ratio of 99.847% of the obtained reads had a phred quality score of over 20. Populations of 94,044 and 111,335 short reads, representing ~0.27% and 0.32% of *G. candidum* Gc6 sRNAs, were mapped to the GcTV2 and GcTV4 genomes from *G. candidum* Gc6, respectively. A minority of the assembled short reads perfectly matched the corresponding sequences in the totiviral genomes (2097 (2.23%) and 841 (0.75%) reads for GcTV2 and GcTV4, respectively). Mostly, the reads had a single mismatch to the genomes of either virus. Some reports indicate that mismatches in siRNA prevent gene silencing while others report that this can be tolerated [58,59]. The profile of siRNA mutations against three plant viruses was revealed and found to be due to viral or cellular RdRps [60]. Unlike viral-derived sRNAs, approximately 23% (7,630,682) of the short reads mapped to the fungal reference genome perfectly matched their corresponding sequences. This confirms that the prevalence of virus-derived short reads with one mismatch is not due to sequencing errors.

Diverse types of sRNAs have been identified in RNA-virus-infected hosts such as siRNAs, unusually small RNAs (usRNAs), and Piwi-interacting RNAs (piRNAs) [61,62]. Virus-derived usRNAs range in length from 13 to 19 nt and were abundantly identified among sRNA populations of human and viral origin [63,64]. usRNAs are shorter than the canonical length of functional miRNAs and are considered neglected intermediate degradation molecules. However, previous reports indicate that usRNAs derived from miRNA and tRNA could probably function in gene regulation [64]. Among the detected totivirus-derived small RNAs from *G. candidum* Gc6, reads with the common siRNA length (20–24 nt) were not abundant. The longest sRNA segment detected was 20 nt in length, mapped to GcTV2-Gc6, and was represented by only 10 sequences. The rest of the sequenced sRNAs were shorter, ranging from 10 to 19 nt for either virus (Figure 5, Table 1). Therefore, the sequenced totivirus-derived sRNAs from *G. candidum* Gc6 may represent populations of usRNAs, mostly with a single mismatch in their sequence. To eliminate the possibility that the failure to detect viral-derived sRNAs with the common siRNA length was not due to sequencing issues, comparisons were made between the number and length of viral-derived and host-derived sRNAs (Table 1). About 51.5% of the host-derived reads ranged in length between 20 and 24 nt, confirming the validity of the sequencing process and data obtained for the viral usRNAs. 

Although it was clearly shown that RNA silencing is likely active in *G. candidum* isolate Gc6, the RNAi pathway is absent in some eukaryotes including *Saccharomyces cerevisiae*. Studies have shown that the loss of the RNAi machinery during evolution is beneficial to the yeast fungi as this helps in maintaining the killer virus system in the host fungus. Experimental restoration of the RNAi pathway caused a decreased ability to maintain the killer virus system, resulting in increased sensitivity to toxins from RNAi-deficient killer virus-containing cells [65]. Similarly, scarcity of viral-derived siRNAs with a functional length in *G. candidum* isolate Gc6 might be a result of an evolutionary mechanism to lose the RNAi pathway against viral infection and hence beneficial to the host fungi.

In summary, the inability to detect siRNAs, the abundance of usRNAs with mismatches, and the successful accumulation of totiviral genomes as shown by the dsRNA and electrophoresis data suggests that the host–defense machinery is not fully able to suppress the replication of *G. candidum* viruses and/or the totiviruses manage to counteract the RNAi mechanisms by expressing viral suppressors of RNAi (VSRs). Although VSR expression has been previously reported for several mycoviruses [57], our speculations regarding the ability of *G. candidum* viruses to express VSRs and the functional viability of their usRNAs need to be confirmed in future studies.

## 4. Conclusions

In this study, two members of the family *Totiviridae* were characterized from the commensal yeast *G. candidum*. Their genome organization is similar to that of many other totiviruses whose genes have the potential to be expressed as a fusion protein through a −1 frameshifting mechanism. A phylogenetic analysis placed the two viruses into two different sub-clades of the *Totivirus* genus. To our knowledge, this is the first report of mycovirus-derived usRNAs. The *G. candidum*–totivirus system may provide an easy model to study the role of usRNAs in fungi and other organisms. It can also be used to further reveal if *G. candidum* is deficient in RNAi as is the case for *S. cerevisiae*. 

## Figures and Tables

**Figure 1 viruses-15-02150-f001:**
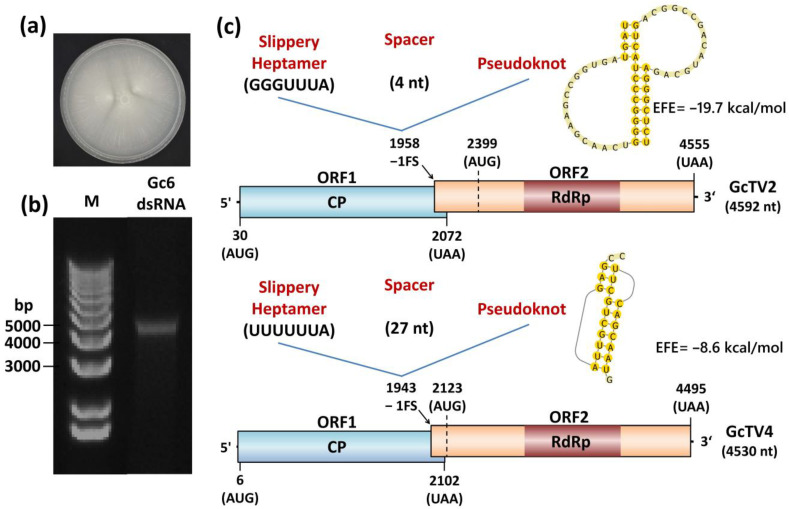
(**a**) Colony morphology of *Geotrichum candidum* isolate Gc6 grown on PDA media. (**b**) Agarose gel electrophoresis of dsRNAs purified from *G. candidum* Gc6. Purified dsRNA fraction was treated with DNase and RNase in a high salt buffer before electrophoretic separation. M: 1 kb plus DNA marker (Invitrogen). (**c**) Schematic representation of the genome organization of *G. candidum* Gc6 totiviruses. Coding regions are indicated by colored boxes whereas untranslated regions are indicated by short lines at both termini. The potential −1 frameshifting site (−1FS) is indicated by black arrows and its three components (slippery heptamer, spacer, and pseudoknot) are shown as an inset in each totiviral genome schematic representation. EFE refers to estimated free energy values for the H-type pseudoknots.

**Figure 2 viruses-15-02150-f002:**
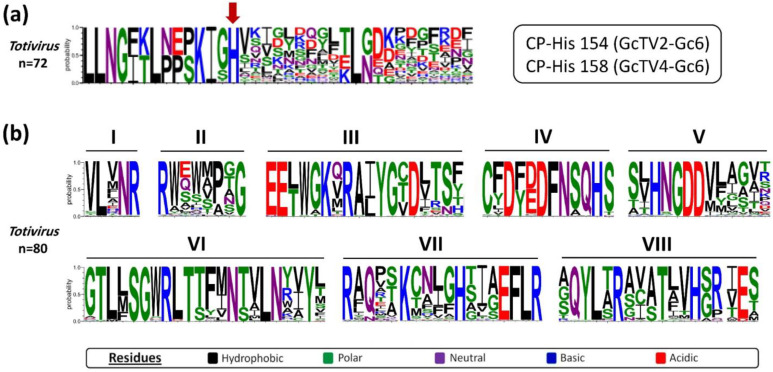
(**a**) Partial multiple amino acid (aa) sequence alignment of the CP sequences of totiviruses including those of *G. candidum*. The conserved histidine residue required for cap snatching is indicated by a red arrow. (**b**) aa sequence alignments showing the conserved motifs (I–VIII) of RNA-dependent RNA polymerase (RdRp) sequences of *G. candidum* totiviruses and other members of the genus *Totivirus*.

**Figure 3 viruses-15-02150-f003:**
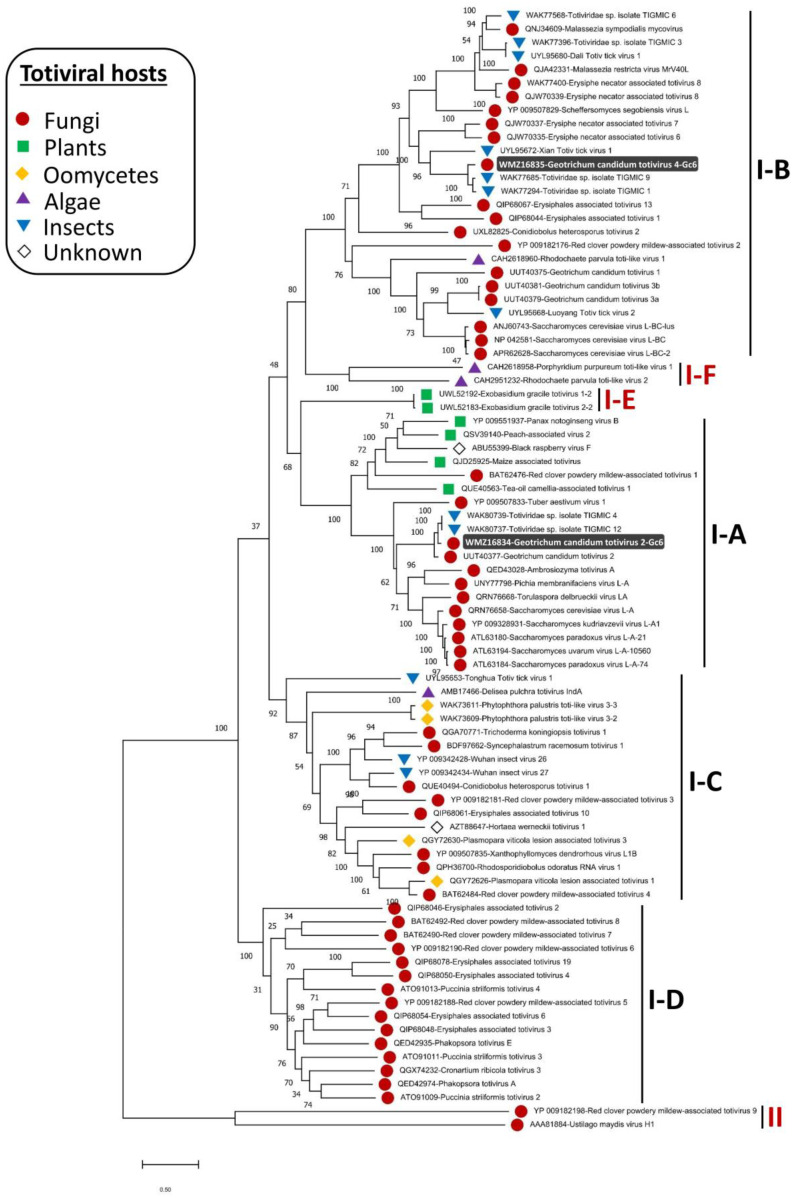
Maximum likelihood phylogenetic tree based on multiple alignments of RNA-dependent RNA polymerase amino acid (aa) sequences of Geotrichum candidum totivirus 2 (GcTV2-Gc6), GcTV4-Gc6, and other members of the genus *Totivirus*. The phylogenetic tree was constructed using MEGA-X software and LG + G + I as the best evolutionary model with 1000 bootstrap replicates.

**Figure 4 viruses-15-02150-f004:**
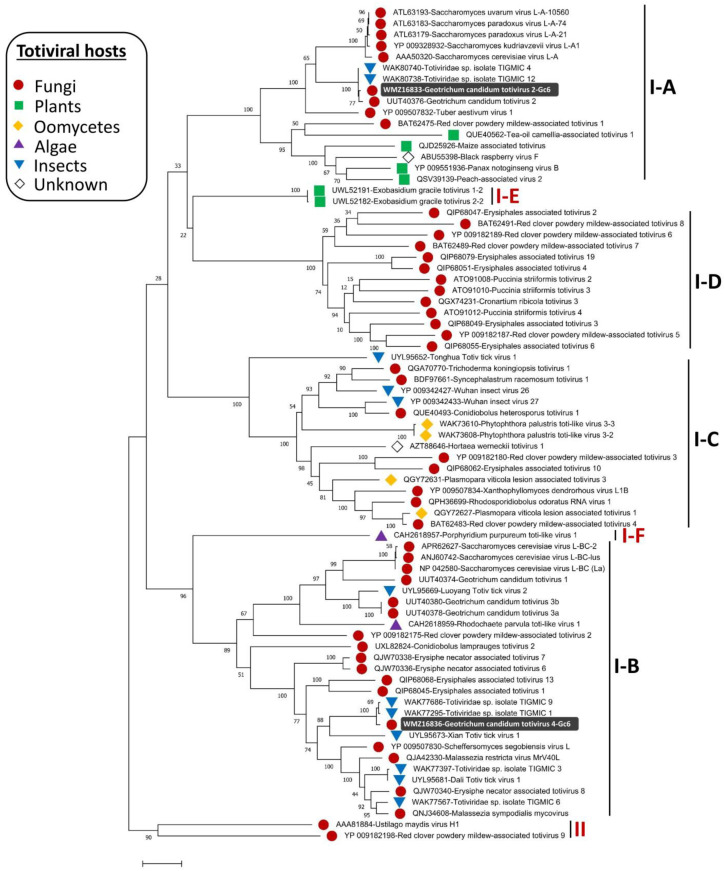
Maximum likelihood phylogenetic tree based on multiple alignments of capsid protein (CP) amino acid (aa) sequences of Geotrichum candidum totivirus 2 (GcTV2-Gc6), GcTV4-Gc6, and other members of the genus *Totivirus*. The phylogenetic tree was constructed using MEGA-X software and rtREV + *G* + *F* as the best evolutionary model with 1000 bootstrap replicates.

**Figure 5 viruses-15-02150-f005:**
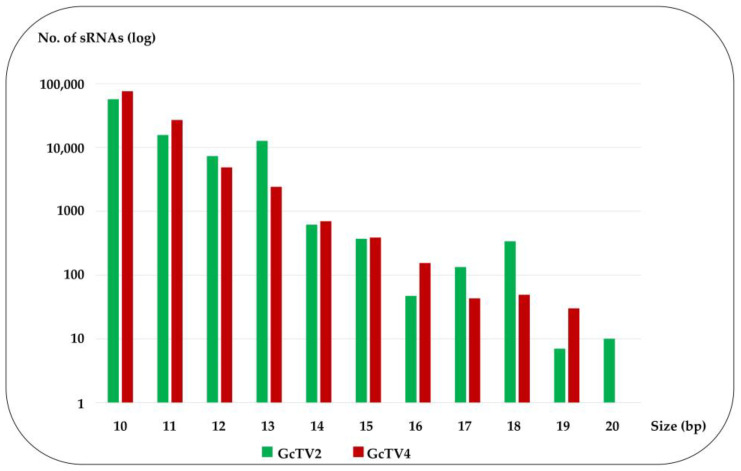
Length distribution in log scale of small RNAs (sRNAs) mapped against Geotrichum candidum totivirus 2 (GcTV2-Gc6) and GcTV4-Gc6.

**Table 1 viruses-15-02150-t001:** Size distribution of small RNA (sRNA) sequencing reads mapped to the genomes of GcTV2-Gc6, GcTV4-Gc6, and *G. candidum* reference genome LMA-244_clib (GenBank: GCA_013365045.1). Numbers of sRNA reads with 0 and 1 mismatches are presented. ND: not detected.

sRNA Length (nts)	Total Viral		Total Fungal	Perfect Match			1 Mismatch		
	GcTV2	GcTV4		GcTV2	GcTV4	*G. candidum*	GcTV2	GcTV4	*G. candidum*
10	56,987	75,858	579,036	1450	545	579,020	55,537	75,313	16
11	15,616	26,851	439,532	522	232	436,820	15,094	26,619	2712
12	7309	4859	980,610	99	43	866,277	7210	4816	114,333
13	12,614	2410	1,247,012	21	8	646,920	12,593	2402	600,092
14	615	695	2,423,441	5	8	257,021	610	687	2,166,420
15	369	386	2,374,600	ND	5	243,264	369	381	2,131,336
16	47	154	1,670,491	ND	ND	262,674	47	154	1,407,817
17	133	43	3,120,220	ND	ND	301,826	133	43	2,818,394
18	337	49	1,246,059	ND	ND	337,262	337	49	908,797
19	7	30	2,103,211	ND	ND	602,221	7	30	1,500,990
20	10	ND	12,694,084	ND	ND	1,312,581	10	ND	11,381,503
21	ND	ND	1,503,777	ND	ND	535,043	ND	ND	968,734
22	ND	ND	1,025,590	ND	ND	351,560	ND	ND	674,030
23	ND	ND	1,181,768	ND	ND	269,011	ND	ND	912,757
24	ND	ND	1,518,337	ND	ND	233,035	ND	ND	1,285,302
25	ND	ND	240,986	ND	ND	121,743	ND	ND	119,243
26	ND	ND	151,291	ND	ND	79,791	ND	ND	71,500
27	ND	ND	109,991	ND	ND	64,998	ND	ND	44,993
28	ND	ND	87,651	ND	ND	51,255	ND	ND	36,396
29	ND	ND	49,420	ND	ND	23,245	ND	ND	26,175
30	ND	ND	39,603	ND	ND	19,653	ND	ND	19,950
>30	ND	ND	77,785	ND	ND	35,462	ND	ND	42,323
Total	94,044	111,335	34,864,495	2097	841	7,630,682	91,947	110,494	27,233,813

## Data Availability

GcTV2 and GcTV4 genome sequences are available in GenBank under the accession numbers OR250782 and OR250783, respectively.

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
