# Peer review of "Molecular Characterization of Two Totiviruses from the Commensal Yeast Geotrichum candidum"

_viruses, 2023, doi:10.3390/v15112150_

Round 1
Reviewer 1 Report
Comments and Suggestions for Authors
The manuscript submitted by Khalifa et al. describes the discovery of a new totivirus in Geotrichum candidum isolated from soil. The manuscript is well-written, an the figures are clearly presented by the authors. The manuscript details the genetic organization of the two viruses that are found to co-infect the same isolate of fungi, their genetic organization, and phylogenetic relationship to other totiviruses that have been described. The article is straightforward and does not make any claims that are not supported by the presented data, although there are some data that have not been fully presented and would need to be included. My comments on the manuscript are minor corrections and suggested additions to the existing datasets.
Line 16 – incomplete sentence and parenthesis.
Line 22 – “It is interesting that all viruses identified from G. candidum belong to the genus Totivirus.” Why? This is an empty open-ended statement that doesn’t fit well in the abstract. It doesn’t add any information on the study to the reader.
Line 34 – Justfiy the use of reference one. It doesn’t directly discuss the role of Geotrichum candidum in supporting the growth of Pencillium camemberti. There are other more direct references that should be used.
Line 85 “T4L-ligated dsRNAs purified” How were they purified?
Line 89 – “and sequenced by” – what method was used?
Line 110 – “Fungal viruses, those able to replicate in fungi,” When would a fungal virus not be able to replicate in a fungus? Reword for clarity.
Line 110- 121. This section repeats a lot of what is already said in the intro. This text does not fit well under the title of “results and discussion”. Consider removing.
Line 130 – “greater than 98% with” – provide a phylogeny of the fungal ITS regions to confirm the assignment of isolate Gc6 as G. candidum.
Line 136 – Figure 1A – this should be refered to as Figure 1B. Figure 1A (is not referred to in the text.
Figure 1A – I am not sure what the authors are trying to communicate with this panel. Is there something particular about the growth of thisorganism that is noteworthy? Perhaps some controls are needed – other isolates of the same species for comparison?
Figure 1B – The methods and results need to clarify whether the dsRNAs shown in this figure have been treated with DNase and RNase before running on the agarose gel. The legend of the figure suggests that this sample is dsRNAs purified from G. candidum, with no mention of enzymatic treatment. The authors should clarify.
Figure 1C – There are several abbreviations that need to be defined in the legend – FS, EFE.
Line 136 – as mentioned above it is unclear where the DNAse and RNAse digestion data is presented.
Line 145 and 149 – Provide the BlastX results in supplement. Which viruses were the closes match?
Line 147 – RACE sequencing was not explicitly mentioned in the methods section.
Line 164 – Is there any common nucleotide sequence motif of these viruses with other Totiviruses?
Figure 3 – Can the authors better highlight the viruses that they have discovered in the phylogeny?
Figure 3 – there are more yeast viruses that would be useful reference points on the RdRP phylogeny for Geotrichum candidum totivirus 2-Gc6 - Torulaspora delbrueckii virus LA, Ambrosiozyma totivirus A, and Pichia membranifaciens virus L-A.
Figure S1 – Why do the authors not also present a phylogeny for the capsid protein in the main paper?
Line 273 – “A minority of the assembled short reads perfectly matched the corresponding sequences in totiviral genomes. Mostly, the reads had a single mismatch to the genomes of either virus.” Can the author please create a graph to show the distribution of sRNAs with 0, 1, 2, 3 etc mismatches? Perhaps correlated with the size of these sRNAs? In the tex it would be better to have actual numbers for the proportion of mismatched sRNAs rather than just saying “mostly”.
Figure 4 – A log scale would be more useful to show the low numbers of some of the sRNAs.
Author Response
Reviewer 1
- Line 16 – incomplete sentence and parenthesis.
The sentence has been corrected.
- Line 22 – “It is interesting that all viruses identified from G. candidum belong to the genus Totivirus.” Why? This is an empty open-ended statement that doesn’t fit well in the abstract. It doesn’t add any information on the study to the reader.
This can probably be due to the lack of research looking at mycoviruses from this host. Future research from this host may add novel mycoviruses from other known or newly established families. We have extended the sentence “however this might be due to the lack of research reporting the characterization of mycoviruses from this fungal host” to clarify this point.
- Line 34 – Justfiy the use of reference one. It doesn’t directly discuss the role of Geotrichum candidum in supporting the growth ofPencillium camemberti. There are other more direct references that should be used.
We agree. Two references have been added instead of reference 1 and the references list has been updated.
- Line 85 “T4L-ligated dsRNAs purified” How were they purified?
Adapter-ligated dsRNAs were purified using a PCR clean-up & gel extraction kit (GeneDireX, Taiwan). This information has been added to the text.
- Line 89 – “and sequenced by” – what method was used?
The terminal sequences were obtained using Sanger sequencing. This information has been added to the manuscript.
- Line 110 – “Fungal viruses, those able to replicate in fungi,” When would a fungal virus not be able to replicate in a fungus? Reword for clarity.
This was to differentiate between fungal viruses (replicative) and viruses present within the fungal vectors during transmission (i.e. plant viruses).
- Line 110- 121. This section repeats a lot of what is already said in the intro. This text does not fit well under the title of “results and discussion”. Consider removing.
This section was modified in the introduction and shortened in the results and discussion as suggested.
- Line 130 – “greater than 98% with” – provide a phylogeny of the fungal ITS regions to confirm the assignment of isolate Gc6 as candidum.
A phylogenetic tree was added as a supplementary figure (S1).
- Line 136 – Figure 1A – this should be refered to as Figure 1B. Figure 1A (is not referred to in the text.
This was corrected and referred to as “Figure 1b”. Figure 1a was also mentioned in the text accordingly.
- Figure 1A – I am not sure what the authors are trying to communicate with this panel. Is there something particular about the growth of this organism that is noteworthy? Perhaps some controls are needed – other isolates of the same species for comparison?
This figure 1a was included to show a pure culture of isolate Gc6. Unfortunately, we have not yet obtained other virus-free isogeneic isolates to make comparisons of cultural characteristics.
- Figure 1B – The methods and results need to clarify whether the dsRNAs shown in this figure have been treated with DNase and RNase before running on the agarose gel. The legend of the figure suggests that this sample is dsRNAs purified from G. candidum, with no mention of enzymatic treatment. The authors should clarify.
It is already mentioned in the materials & methods and results section that the dsRNA was treated with DNase and RNase at high salt buffer concentration before running on agarose gel. As suggested by the reviewer, this point was clarified in the figure legend.
- Figure 1C – There are several abbreviations that need to be defined in the legend – FS, EFE.
Thanks for the note. The mentioned abbreviations were defined. - Line 136 – as mentioned above it is unclear where the DNAse and RNAse digestion data is presented.
Thanks for your comment. This has been clarified and as previously mentioned, treatment was accomplished before running on agarose gel.
- Line 145 and 149 – Provide the BlastX results in supplement. Which viruses were the closes match?
Blastx results were presented as supplementary tables (Tables S2 and S3). Hits were arranged based on their identity to GcTV2 and GcTV4.
- Line 147 – RACE sequencing was not explicitly mentioned in the methods section.
Adapter ligation and sequencing the dsRNA terminal sequences is detailed at Line 85.
- Line 164 – Is there any common nucleotide sequence motif of these viruses with other Totiviruses?
Multiple nucleotide sequence alignments showed that there are regions with high nucleotide sequence similarities (not identical). The regions with the highest similarities are located approximately from nt positions 2800 to 4000 where conserved motifs of RdRp are found.
- Figure 3 – Can the authors better highlight the viruses that they have discovered in the phylogeny?
Completed as requested. Virus names were highlighted in the phylogenetic trees of the revised version.
- Figure 3 – there are more yeast viruses that would be useful reference points on the RdRP phylogeny for Geotrichum candidum totivirus 2-Gc6 - Torulaspora delbrueckii virus LA, Ambrosiozyma totivirus A, and Pichia membranifaciens virus L-A.
Completed as requested. These yeast viruses were added to the phylogenetic tree.
- Figure S1 – Why do the authors not also present a phylogeny for the capsid protein in the main paper?
The CP phylogenetic tree was presented in the main text (Figure 4) as suggested.
- Line 273 – “A minority of the assembled short reads perfectly matched the corresponding sequences in totiviral genomes. Mostly, the reads had a single mismatch to the genomes of either virus.” Can the author please create a graph to show the distribution of sRNAs with 0, 1, 2, 3 etc mismatches? Perhaps correlated with the size of these sRNAs? In the tex it would be better to have actual numbers for the proportion of mismatched sRNAs rather than just saying “mostly”.
The distribution of sRNAs with/without mismatches was presented in Table 2. Proportions were presented in the main text as well as the table. Moreover, sRNAs were also mapped to G. candidum reference genome as a control for the validity of sRNA extraction and sequencing.
- Figure 4 – A log scale would be more useful to show the low numbers of some of the sRNAs.
Completed as requested. Logarithmic scale was used in the revised Figure 5.
Reviewer 2 Report
Comments and Suggestions for Authors
Manuscript number viruses-2636151, titled “Molecular characterization of two totiviruses from the commensal yeast Geotrichum candidum”.
Using high throughput sequencing, Khalifa et al. detected two co–infected totiviruses in the commensal yeast Geotrichum candidum, with one known and one novel. Furthermore, they cloned the full-length and characterized their molecular traits of both totiviruses, as well as their related small inference RNAs. This manuscript fits in the scope of Viruses, but contribute less knowledge to the field considering that no biological traits related to both totiviruses were characterized. I suggested to reject it unless their biological traits are further clarified.
Other comments:
1) The presentation should be improved, e.g., the first paragraph of the section “Results and discussion” in lines 109-121 should be moved to introduction section.
2) There are many English mistakes should be improved, see the pdf that was marked by me.

Please find the attached pdf version that was marked or changed.
Author Response
Reviewer 2
1) The presentation should be improved, e.g., the first paragraph of the section “Results and discussion” in lines 109-121 should be moved to introduction section.
Completed as requested. This section was modified in the introduction and shortened in the results and discussion as suggested.
2) There are many English mistakes should be improved, see the pdf that was marked by me.
Language issues were corrected as appropriate.
Reviewer 3 Report
Comments and Suggestions for Authors
This manuscript describes the characterization of novel totiviruses from the yeast fungus Geotrichum candidum. In general, the manuscript is well written and the data are well presented. I have several suggestions to further improve the article.
1. In material and method section, it is not clearly described how the yeast isolate was obtained. As many kinds of fungi could be obtained, it is not clear whether the authors particularly target the yeast species during the isolation.
2. Throughout the manuscript, “Isolate Gc6” is used to refer the yeast. It is better to be more specific, for example, “G. candidum Gc6”.
3. Results and discussion Line 110-121, This part is redundant as it has been presented in the introduction section. Should be omitted or shortened.
4. Lane 165-168, This part could be moved to introduction section.
5. In Figure 3, The two novel totiviruses identified in this study should be highlighted or marked, so they are easily to be found.
6. More information should be provided regarding the sRNA data. For examples, how many total numbers of sRNA reads obtained from the data set, how the size distribution of the total sRNA reads, percentage of totivirus-derived sRNAs and the strand polarity of the virus sRNAs.
7. Several fungal species including yeast are known to be deficient in RNAi. Is there any information regarding RNAi machinery in this particular yeast species? This point should be added to the discussion.
Author Response
Reviewer 3
- In material and method section, it is not clearly described how the yeast isolate was obtained. As many kinds of fungi could be obtained, it is not clear whether the authors particularly target the yeast species during the isolation.
Isolate Gc6 was isolated among other fungal species from the soil sample. We did not target yeast species and during isolation, yeast-like and filamentous fungi were obtained. This was followed by obtaining pure cultures for isolates of interest including isolate Gc6.
The word “isolates” was replaced “species” to clarify that yeast fungi were not targeted.
- Throughout the manuscript, “Isolate Gc6” is used to refer the yeast. It is better to be more specific, for example, “G. candidum Gc6”.
Completed as requested.
- Results and discussion Line 110-121, This part is redundant as it has been presented in the introduction section. Should be omitted or shortened.
This section was modified in the introduction and shortened in the results and discussion as suggested.
- Lane 165-168, This part could be moved to introduction section.
We agree that it could be moved to the introduction. However, we meant to present the ideal genome organization of totiviruses before presenting the organization of the two viruses of this study. Therefore, we didn’t move it to the introduction.
- In Figure 3, The two novel totiviruses identified in this study should be highlighted or marked, so they are easily to be found.
Completed as requested. Virus names were highlighted in the phylogenetic trees of the revised version.
- More information should be provided regarding the sRNA data. For examples, how many total numbers of sRNA reads obtained from the data set, how the size distribution of the total sRNA reads, percentage of totivirus-derived sRNAs and the strand polarity of the virus sRNAs.
Completed as requested. We have added as much data as possible to the text and in Table 2.
- Several fungal species including yeast are known to be deficient in RNAi. Is there any information regarding RNAi machinery in this particular yeast species? This point should be added to the discussion.
We couldn’t find information regarding RNAi in G. candidum. However, the loss of RNAi pathway in some yeast fungi and its benefit to the host has been added to the discussion.
Round 2
Reviewer 2 Report
Comments and Suggestions for Authors
The authors did not make further biological characterization as requirment, and I do not think this study is worthy to be published in Virus.
Comments on the Quality of English LanguageThe English usage is good.
Reviewer 3 Report
Comments and Suggestions for Authors
All reviewer's comments have been adressed